# High-mobility group box 1 fragment suppresses adverse post-infarction remodeling by recruiting PDGFRα-positive bone marrow cells

Takasumi Goto[1]*, Shigeru Miyagawa[1], Katsuto Tamai[2], Ryohei Matsuura[1], Takashi Kido[1], Toru Kuratani[3], Kazuo Shimamura[3], Ryoto Sakaniwa[4], Akima Harada[1], Yoshiki Sawa[1]

1 Department of Cardiovascular Surgery, Osaka University Graduate School of Medicine, Osaka, Japan, 2 Department of Stem Cell Therapy Science, Osaka University Graduate School of Medicine, Osaka, Japan, 3 Department of Minimally Invasive Cardiovascular Medicine, Osaka University Graduate School of Medicine, Osaka, Japan, 4 Department of Social Medicine, Osaka University Graduate School of Medicine, Osaka, Japan

* t-goto@surg1.med.osaka-u.ac.jp

**Data Availability Statement:** All relevant data are within the manuscript and its Supporting Information files.

## Abstract

### Objectives

High-mobility group box 1 protein (HMGB1) fragment enhances bone marrow-derived mesenchymal stem cell (BM-MSC) recruitment to damaged tissue to promote tissue regeneration. This study aimed to evaluate whether systemic injection of HMGB1 fragment could promote tissue repair in a rat model of myocardial infarction (MI).

### Methods

HMGB1 (n = 14) or phosphate buffered saline (n = 12, control) was administered to MI rats for 4 days. Cardiac performance and left ventricular remodeling were evaluated using ultrasonography and immunostaining. BM-MSC recruitment to damaged tissue in green fluorescent protein-bone marrow transplantation (GFP-BMT) models was evaluated using immunostaining.

### Results

At four weeks post-treatment, the left ventricular ejection fraction was significantly improved in the HMGB1 group compared to that in the control. Interstitial fibrosis and cardiomyocyte hypertrophy were also significantly attenuated in the HMGB1 group compared to the control. In the peri-infarction area, VEGF-A mRNA expression was significantly higher and TGFβ expression was significantly attenuated in the HMGB1 group than in the control. In GFP-BMT rats, GFP$^+$/PDGFRα$^+$ cells were significantly mobilized to the peri-infarction area in the HMGB1 group compared to that in the control, leading to the formation of new vasculature. In addition, intravital imaging revealed that more GFP$^+$/PDGFRα$^+$ cells were recruited to the peri-infarction area in the HMGB1 group than in the control 12 h after treatment.

**Funding:** There were no sources of funding for this study.

**Competing interests:** The authors have no competing interests to declare.

## Conclusions

Systemic administration of HMGB1 induced angiogenesis and reduced fibrosis by recruiting PDGFRα+ mesenchymal cells from the bone marrow, suggesting that HMGB1 administration might be a new therapeutic approach for heart failure after MI.

## Introduction

Although the rate of mortality owing to acute myocardial infarction (MI) has decreased considerably in recent years concomitant with the evolution of coronary reperfusion interventions, MI remains one of the leading causes of chronic heart failure (CHF) that occurs as a consequence of adverse left ventricular (LV) remodeling [1, 2]. In the past decade, there has been significant progress in regenerative therapies using different types of stem cells for suppressing adverse LV remodeling.

In the human body, there are several endogenous regenerative mechanisms that function in the repair of injured organs. Bone marrow-derived mesenchymal stem cells (BM-MSCs) or tissue stem cells play an important role in repairing damaged tissues [3, 4]. BM-MSCs, in particular, show a strong potential to repair damaged organs by recruiting other host cells, secreting different growth factors, or differentiating into various cells, such as endothelial cells [5], thereby promoting angiogenesis and inhibiting adverse fibrosis in various injuries and diseases including injured muscles [6], cerebral infarction [7], and MI [8, 9]. As an alternative to conventional cellular therapy, by enhancing endogenous repair mechanisms with stem cells or pharmacological agents, or by accelerating endogenous regenerative function, could offer a less invasive regenerative treatment.

High-mobility group box 1 protein (HMGB1), a non-histone nuclear protein regulating chromatin structure [10], has two different effects as follows: HMGB1 released from necrotic cells activates macrophages and neutrophils, thereby accelerating inflammation in injured tissues [10]; in contrast, HMGB1 is also a regenerative factor that enhances the mobilization of PDGFRα+ mesenchymal cells from the bone marrow to damaged tissue [11–13]. PDGFRα is a main marker of MSCs in bone marrow [14]. The inflammatory reactions elicited by HMGB1 are induced via binding between specific HMGB1 domains and Toll-like receptors-2/-4 or the receptor for advanced glycation end product [15, 16]. By resecting the previous reported functional domains of HMGB1 associated with inflammatory reactions, we have created a novel HMGB1 fragment (Fig 1). We previously determined that this HMGB1 fragment is associated with the inhibition of adverse ventricular remodeling [11].

In the present study, we used a rat model of MI to test the hypothesis that systemic administration of this HMGB1 fragment promotes tissue regeneration by mobilizing BM-MSCs to the damaged heart tissue.

## Materials and methods

### Animal care

All experimental procedures and protocols were approved by the institutional ethics committee of Osaka University Graduate School of Medicine (approval number 28-024-02). Animal care was reviewed and approved by the National Institutes of Health Publication, "Guide for the Care and Use of Laboratory Animals". All animal procedures followed in the present study conform to the guidelines from Directive 2010/63/EU of the European Parliament on the protection of animals used for scientific purposes [17]. All experimental animals were euthanized using sufficient isoflurane to minimize animal suffering.

### Short length of HMGB1 fragment

As previously reported [11], the MSC mobilization domain from human HMGB1 was produced as "HMGB1 fragment" by solid-phase synthesis, which is shown in Fig 1. The HMGB1 fragment was provided by StemRIM (Osaka, Japan), and was dissolved in phosphate buffered saline (PBS) to a concentration of 1 mg/ml prior to systemic administration.

### Examination 1: CHF after MI with administration of systemic HMGB1 fragment

For the study protocol of the first examination, seven-week-old male Sprague-Dawley rats (SD rats; 200–250 g, Oriental Yeast Co. ltd, Japan) underwent left thoracotomy under 1.5% isoflurane inhalation and mechanical ventilation using a volume-controlled ventilator [18]. The left coronary artery (LCA) was ligated with 6–0 polypropylene (C-1, ETHICON) at the level of the bottom edge of the left atrial appendage. At 2 weeks after LCA high ligation, rats were divided into two subgroups and received either HMGB1 (3 mL/kg/day; n = 14) or the same volume of PBS (3 mL/kg/day; n = 12) as a control by injection into the femoral veins for 4 consecutive days under 1.5% isoflurane inhalation via a nose cone. The dose of HMGB1 was based on our previous studies [11–13]. At 4 weeks after HMGB1 treatment, LV adverse remodeling was assessed using histological and real-time PCR (RT-PCR) analyses.

### Examination 2: BM-MSC mobilizing factor in a rat model of MI

To evaluate the homing factor of MSCs in the damaged heart tissue prior to HMGB1 treatment, the second examination was performed. We focused on stromal cell derived-factor-1 (SDF1), a well-known homing factor of MSCs [19, 20]. Using the MI model rat described above (n = 6), SDF1 expression at peri-infarction zone, ventricular septal zone, and remote zone was compared to those in normal rats of the same age (n = 10) by histological analysis and RT-PCR analysis. Histology of the peri-infarction area was investigated by evaluating vascular endothelial cells using electron microscopy.

### Examination 3: Generation of green fluorescent protein-bone marrow transplantation (GFP-BMT) model

To investigate whether BM-MSCs would be mobilized from the bone marrow to the MI lesion, a third examination using the GFP+ BM transplantation (BMT) rat MI model was performed.

For the study protocol of the third examination, BM cells were isolated from 9- to 11-week old male Wistar (W)-Tg (CAG-GFP) 184Ys rats that ubiquitously expressing enhanced GFP (National BioResource Project-rat, Japan). Isolated GFP-BM cells ($3 \times 10^7$) were injected via the tail vein into five-week old male Wistar rats (CLEA Japan. Inc, Japan) that had received lethal irradiation of 10 Gy of X-rays [21]. Five weeks after GFP+ BM transplantation, LCA ligation

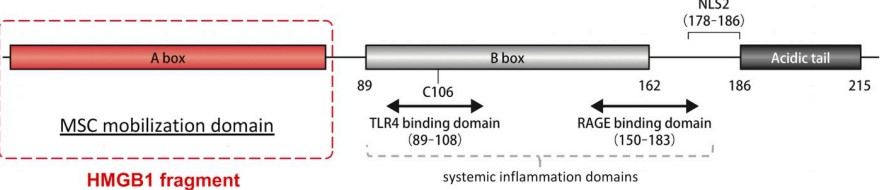

**Fig 1. Schematic of the HMGB1 fragment.** A novel domain of HMGB1 is associated with enhancing the mobilization of BM-MSCs (Red lines). Recombinant HMGB1 fragment has been created using this domain.

was performed. Two weeks after LCA ligation, HMGB1 (n = 8) or PBS (n = 7) was administered for 4 consecutive days as in the first examination. Four weeks after each treatment, GFP expression in the peri-infarction area was evaluated in each group using RT-PCR analysis. The recruitment of GFP+ BM-MSCs to the damaged heart tissue was assessed using immuno-histological analysis. We also investigated the therapeutic effects of the recruited BM-MSCs associated with paracrine activity and differentiation into the vessel constituent cells using immuno-histological analysis.

## Protocol of intra-vital imaging using two-photon microscopy

As significant recruitment of BM-MSCs to blood circulation was observed at 12 h after systemic HMGB1 injection in our previous study [12], intra-vital imaging was performed as previously described to evaluate the mobilization of BM-cells to damaged myocardium in real time [22]. The GFP+-BMT rat MI model was placed on a stainless plate under 1.5% isoflurane inhalation and mechanical ventilation using a volume-controlled ventilator. After general anesthesia, a central venous catheter (CVC) was inserted through the femoral vein. Thereafter, the chest wall was resected using bipolar scissors (Force FX-CS, E4051CT; Valleylab, Denver, USA), and the peri-infarction area of the beating heart was affixed by suction to the central aperture of an original stabilizer. The intra-vital microscope system comprised a two-photon microscope (A1-MP; Nikon, Japan) incorporating a laser (Chameleon Vision II Ti:Sapphire; Coherent, Santa Clara, CA, USA) tuned to 800–880 nm and an upright microscope equipped with a 25× water immersion objective lens (CFI Apo 25 × W MP; Nikon). An original plate-stabilizer setup was placed on two-axis translation stage under the objective lens in a temperature-controlled dark box. After adjusting the objective lens through the bioscope, the beating heart was viewed in vivo via the central aperture. Isolectin-B4 was injected via the CVC, and HMGB1 (n = 6) or PBS (n = 5) was injected at the same dose as used in the first examination. After each administration, continuous live imaging was performed over the subsequent 12 h. As in the third examination, the average number of GFP+/PDGFRα+ cells at the peri-infarction area was assessed using histological analysis.

## Echocardiography

Cardiac functions were evaluated by echocardiogram using SONOS-7500 and s12-probe (89 Hz, PHILIPS, Netherlands) under 1.5% isoflurane inhalation through a nose cone. Left ventricular diastolic and systolic diameter (LVDd and LVDs) were obtained from parasternal short-axis views. Additionally, left ventricular ejection fraction (LVEF) was calculated from these diameters. In the first examination, these parameters were assessed before HMGB1 treatment, and were measured again at 1 and 4 weeks after HMGB1 treatment. In the third examination, LVEF was evaluated prior to HMGB1 treatment and remeasured at 4 weeks after HMGB1 treatment.

## Histological analysis

The heart was resected perpendicularly to the long axis of the left ventricle in slices of a few mm. All the excised heart specimens were fixed with 10% buffered formalin for paraffin-embedded sections or 4% paraformaldehyde for frozen sections for over a day.

In the first examination, paraffin-embedded sections were resected in 2-μm slices and stained with hematoxylin-eosin (HE) and picrosirius-red to evaluate the fibrotic area using light microscopy (Leica, Wetzlar, Germany). The paraffin-embedded sections were also stained with periodic acid-Schiff (PAS) to assess cardiomyocyte hypertrophy in each group. Using light microscopy with BZ-analysis software (Keyence, Tokyo, Japan), the short-axis

diameter of myocytes was counted in 10 randomly selected fields, and the average number calculated. Neovascularization in the peri-infarction area was evaluated using rabbit anti-von Willebrand factor polyclonal antibody (1:200; Dako, Glostrup, Denmark). Using light microscopy with BZ-analysis software, the number of von Willebrand factor-positive cells in 10 randomly selected fields was counted, and the average number calculated. MSCs mobilization to the damaged myocardium was evaluated using double staining with mouse anti-CD90 monoclonal (1:100; Abcam, Cambridge, UK) and rabbit anti-PDGFRα polyclonal (1:100; Thermo Fisher Scientific, Waltham, USA) antibodies for each group. Ten different fields at the peri-infarction area were randomly selected, and CD90+/PDGFRα+ cells were counted using confocal laser microscopy (Olympus, FV1000-D IX81, Tokyo, Japan). In the second examination, the frozen sections in MI and normal rats were stained with rabbit anti-SDF1 polyclonal antibody (1:50; Abcam, Cambridge, UK), and were evaluated using the confocal laser microscopy. In the third examination, the frozen sections were stained with PDGFRα (1:100; Thermo Fisher Scientific, Waltham, USA). Ten different fields of the peri-infarction area were randomly selected in each group, and GFP+/PDGFRα+ cells were counted using confocal laser microscopy. The average number was calculated in each group. Paracrine activity of BM-MSCs was assessed with antibody staining for vascular endothelial growth factor A (VEGF-A: rabbit anti-VEGF-A polyclonal antibody, 1:100; Abcam, Cambridge UK). Differentiation of the recruited BM-MSCs was assessed with staining of vascular endothelial cells using isolectin-B4 Alexa Fluor™ 568 conjugate (1:200; Thermo Fisher Scientific, Waltham, USA), or pericytes with rabbit anti-NG2 polyclonal antibody (1:100; LifeSpan Biosciences, Washington, USA).

## Real-time PCR analysis

Real-time PCR was performed as previously described [11]. Total RNA was extracted from cardiac tissue (peri-infarction area, ventricular septum, and remote area) and reverse transcribed using TaqMan reverse transcription reagents (Applied Biosystems, Stockholm, Sweden). RT-PCR was performed on an ABI PRISM 7700 system (Applied Biosystems) using rat-specific primers for *VEGF-A*, *transforming growth factor-β (TGFβ)*, *interleukin (IL)-1β*, and *IL-6* (Assay ID: Rn01511601_m1, Rn00572010_m1, Rn00580432_m1, and Rn01410330_m1, respectively) in the first examination; for *SDF1* (Assay ID: Rn00573260_m1) in the second examination; and for *GFP* (Assay ID: Mr04097229_mr) in the third examination. Each cDNA sample was evaluated in duplicate. Expression of target genes was normalized to that of glyceraldehyde-3-phosphate dehydrogenase (GAPDH) for each sample. Relative gene expression was determined using the $2^{-\Delta\Delta C_T}$ method.

## Statistical analysis

All statistical analyses were performed using JMP Pro 13.0.0 (SAS Institute, Inc., Cary, NC, USA). All data were expressed as the mean ± standard deviation (SD). Between-group differences were compared using the Welch's t-test. A *P*-value (*P*) < 0.05 was considered statistically significant.

## Results

### Examination 1: Improvement of cardiac function after HMGB1 treatment

Details of the first examination are listed in Fig 2A. Standard transthoracic echocardiogram was performed at 0, 1 and 4 weeks after each injection (Fig 2B). Regarding baseline level of LVEF (HMGB1 group vs. control; 42.25 ± 5.07% vs. 43.94 ± 4.89%, *P* = 0.40), LVDd (0.968 ± 0.105 vs. 1.013 ± 0.086 mm, *P* = 0.24), and LVDs (0.792 ± 0.103 vs. 0.820 ± 0.087 mm, *P* = 0.45), there

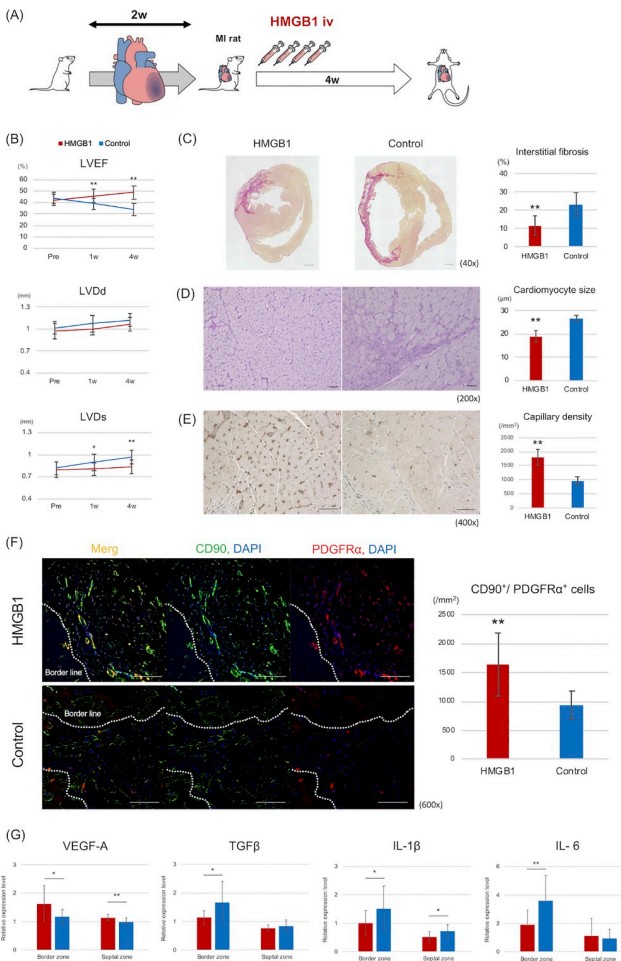

**Fig 2. Evaluation results during the first examination.** The first examination aimed to evaluate the regenerative effect of HMGB1 in a rat model of MI. A: Protocol of first examination. Two weeks after MI, the HMGB1 fragment was administered for 4 days. Four weeks after HMGB1 fragment treatment, histological analyses were performed. B: Echocardiogram revealed that LVEF was significantly higher in the HMGB1 group (n = 14) than in the control (n = 12), at 4 weeks after each treatment. LVDs was significantly shorter in the HMGB1 group than in the control. C-E: LV adverse remodeling in each group was assessed by histological analysis. Interstitial fibrosis was assessed by Picrosirius-red staining (C. representative photomicrographs, 40×, scale bar = 1 mm). Fibrosis was significantly attenuated in the HMGB1 group compared with that in the control. Cardiomyocyte hypertrophy was assessed by Periodic acid-Schiff staining (D. representative photomicrographs, 200×, scale bar = 50 μm). Myocyte size was significantly smaller in the HMGB1 group than in the control. Neovascularization using antihuman von Willebrand factor antibody (E. representative photomicrographs, 400×, scale bar = 50 μm). Capillary density was significantly greater in the HMGB1 group than in the control. F: Evaluation of the recruitment of CD90[+]/PDFGRα[+] cells to the peri-infarction area (600×, scale bar = 50 μm). More CD90[+]/PDFGRα[+] cells were present in the HMGB1 group than in the control. G: RT-PCR analysis was performed in both groups for the following cytokines: *VEGF-A*, *TGFβ*, *IL-1β*, and *IL-6*. *P* -values were calculated using the Welch's t-test. $P < 0.05^*$, $P < 0.01^{**}$.

were no significant differences between each group. LVEF showed greater improvement in the HMGB1 group than in the control at 1 week (45.61 ± 5.926% vs. 39.15 ± 4.908%, *P* = 0.0056), and 4 weeks (48.61 ± 5.51% vs. 33.93 ± 5.27%, *P* < 0.0001) after each administration. LVDs was significantly smaller in the HMGB1 group than in the control at 1 week (0.803 ± 0.091 vs. 0.896 ± 0.110 mm, *P* = 0.027) and 4 weeks later (0.833 ± 0.0905 vs. 0.963 ± 0.095 mm, *P* = 0.0016). Consequently, all rats in each group survived.

## Histological analysis concerning post-MI adverse LV remodeling

Upon histological analysis, interstitial fibrosis was significantly attenuated in the HMGB1 group as compared to the control (fibrotic area; 11.58 ± 5.18% vs. 23.07 ± 6.32%, $P < 0.0001$; Fig 2C). For cardiomyocyte hypertrophy at the peri-infarction area, cardiomyocyte size was significantly smaller in the HMGB1 group than in the control (19.11 ± 2.59 vs. 26.82 ± 1.36 μm, $P < 0.0001$, Fig 2D). Capillary density at the peri-infarction area was significantly greater in the HMGB1 group (1797.98 ± 271.85 vs. 959.04 ± 143.40/mm$^2$, $P < 0.0001$; Fig 2E) than in the control. In addition, comparison of the number of CD90$^+$/PDFGRα$^+$ cells at the peri-infarction area revealed that there were more CD90$^+$/PDFGRα$^+$ cells in the HMGB1 group than in the control (1636.84 ± 538.378 vs. 934.00 ± 250.236/mm$^2$, $P = 0.0003$; Fig 2F).

## Significant increase of VEGF and decrease of TGFβ in HMGB1 group

RT-PCR data for each cytokine expression are shown in Fig 2G. The level of *VEGF-A* mRNA expression in the peri-infarction area was significantly higher in the HMGB1 group than in the control (1.63 ± 0.64 vs. 1.18 ± 0.25, $P = 0.029$). At the septal zone, *VEGF-A* mRNA expression was also significantly higher in the HMGB1 group than in the control (1.14 ± 0.11 vs. 0.99 ± 0.13, $P = 0.0040$). The level of *TGFβ* mRNA expression in the peri-infarction area was significantly lower in the HMGB1 group (1.13 ± 0.25 vs. 1.66 ± 0.75, $P = 0.037$).

With respect to inflammatory cytokines, *IL-1β* mRNA expression at the septal zone was significantly lower in the HMGB1 group than in the control (0.51 ± 0.21 vs. 0.71 ± 0.24, $P = 0.031$). *IL-6* mRNA expression in the peri-infarction area was lower in the HMGB1 group (1.92 ± 1.02 vs. 3.61 ± 1.76, $P = 0.0092$) than in the control.

## Examination 2: Significant increase of SDF1 expression at damaged heart tissue prior to HMGB1 treatment

A second examination was performed to assess the expression of SDF1 in the damaged heart tissue prior to injection of the HMGB1 fragment (Fig 3A). The detailed results are shown in Fig 3. Confocal microscopy imaging showed SDF1 expression along the peri-infarction area in the MI rat. In contrast, there was no significant SDF1 expression in the normal rat (Fig 3B). RT-PCR analyses also showed that *SDF1* mRNA expression in the peri-infarction area was significantly higher in the MI rat than in the normal rat (MI model vs. normal; 2.17 ± 0.48 vs. 0.93 ± 0.16, $P = 0.0010$; Fig 3C). Further, *SDF1* mRNA expression in the border zone was the highest in all areas (septal zone; 1.96 ± 0.96 vs. 1.03 ± 0.25, $P = 0.064$, remote zone; 1.11 ± 0.24 vs. 0.93 ± 0.17; $P = 0.22$).

In electron microscopic analyses of the peri-infarction area, tight junctions between vascular endothelial cells were unclear in MI rats (Fig 4A), and some of the cell-cell junctions were completely destroyed (Fig 4B). In contrast, they were clearly observed in normal rats (Fig 4C and 4D).

## Examination 3: Detection of BM-MSC mobilization to damaged heart tissue by HMGB1 in GFP-BMT model

To investigate whether MSCs could mobilize from the bone marrow to the damaged heart tissue after HMGB1 treatment, a third examination was performed with GFP$^+$ bone marrow transplanted (GFP-BMT) rat MI model (Fig 5A). At 4 weeks after each treatment, a greater improvement in LVEF was observed in the HMGB1 group than in the control, similar to the first examination (49.30 ± 3.75 vs. 36.52 ± 3.09%, $P < 0.0001$; Fig 5B). In RT-PCR analyses, the level of *GFP* mRNA at the peri-infarction area was found to be significantly higher in the

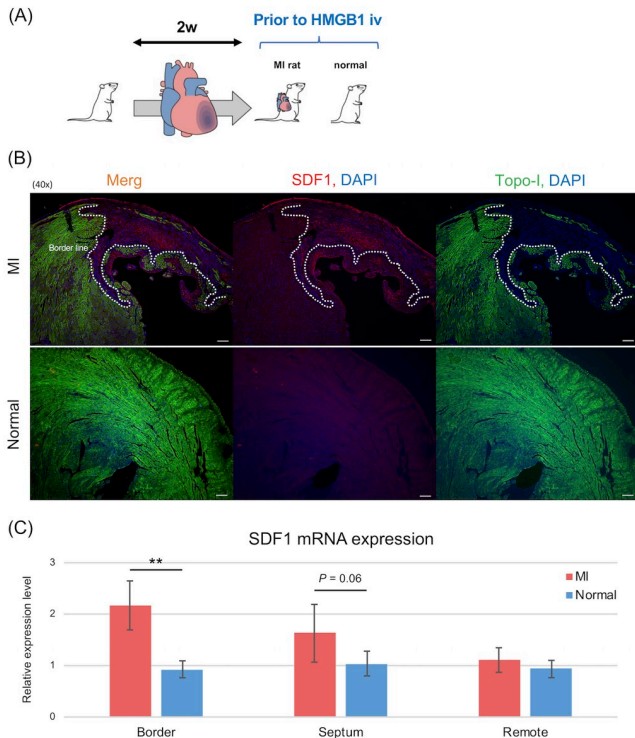

**Fig 3. SDF1 expression at the infarcted heart tissue in MI rats prior to HMGB1 treatment.** The second examination was performed using the same MI rats prior to HMGB1 treatment to assess the expression of SDF1, a representative homing factor of MSCs. A: Study protocol of second examination. B: Histological analysis revealed SDF1 expression along the peri-infarction area in MI model and normal rats (40×, scale bar = 200 μm). C: RT-PCR analysis indicated that SDF1 expression increased significantly in MI rats (n = 6) compared with normal rats (n = 10). *P* -values were calculated using the Welch's t-test. $P < 0.05^*$, $P < 0.01^{**}$.

HMGB1 group than in the control (1.76 ± 0.49 vs. 0.93 ± 0.17, P = 0.017; Fig 5C). Histological analyses also revealed the recruitment of GFP$^+$ cells along the peri-infarction area in the HMGB1 group (Fig 5D–1). Subsequently, we also evaluated GFP$^+$/PDGFRα$^+$ cells in the peri-infarction area and found a significant increase of GFP$^+$/PDGFRα$^+$ cells in the HMGB1 group compared with control (1418.70 ± 243.66 vs. 589.79 ± 66.52/mm$^2$, P < 0.0001; Fig 5D–2). In accord, VEGF paracrine activity of the recruited BM-MSCs was significantly increased after treatment with the HMGB1 fragment in the first examination.

In the HMGB1 group, confocal microscopic imaging showed VEGF-A expression around GFP$^+$/PDGFRα$^+$ cells at the peri-infarction area. In contrast, VEGF-A expression at the border zone was unclear in the control group (Fig 6A). To assess differentiation of the recruited BM-MSCs in the damaged heart tissue, we performed IL-B4 staining and found that some GFP$^+$/PDGFRα$^+$ cells also stained positive for IL-B4. Moreover, some GFP$^+$/PDGFRα$^+$/IL-B4$^+$ cells were present in the vessel at the border zone (Fig 6B). In addition, some GFP$^+$/PDGFRα$^+$/NG-2$^+$ cells also formed part of the vessel in the peri-infarction area (Fig 6C). No GFP$^+$ cardiomyocytes were detected in the present study.

Real time intravital imaging was performed to evaluate GFP$^+$-cells recruitment to damaged myocardium (Fig 7A). The continuous live imaging analyses revealed that GFP$^+$ BM-cells gradually increased after injection with the HMGB1 fragment (Video 1). In contrast, the recruitment of GFP$^+$ BM-cells was not enhanced after PBS injection. At 12 h post-injection with the HMGB1 fragment or PBS, more GFP$^+$/PDGFRα$^+$ cells were visualized in the HMGB1

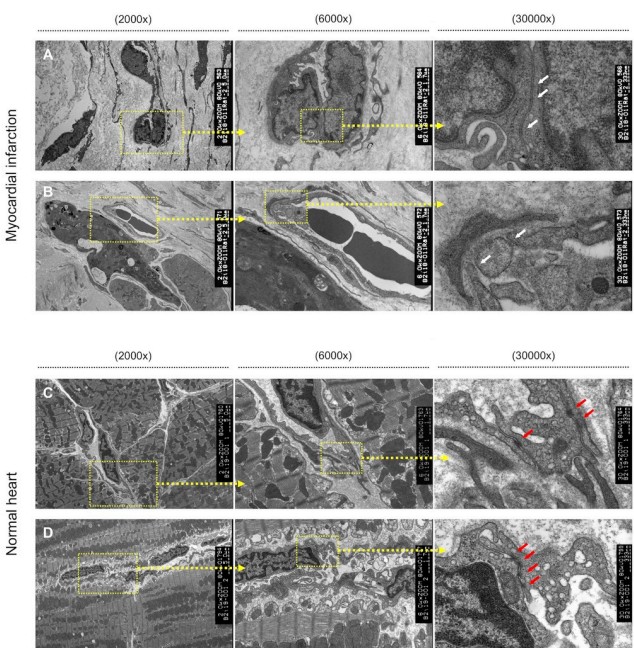

**Fig 4. Electron microscopy analysis of the peri-infarction area prior to HMGB1 treatment.** Representative electron microscopy images (2000×, 6000×, 30000×) of the cell-cell junctions between each vascular endothelial cell in MI rats (A, B) and normal rats (C, D). Whereas those of normal rats were clearly observed (yellow arrows). A: Tight junctions between each vascular endothelial cell are unclear in MI rats (white arrows). B: In the peri-infarction area, some of the cell-cell junctions are destroyed and detached (white arrows). C, D: Electron microscopy analysis of a normal rat heart. The tight junctions between each vascular endothelial cell are clearly observed (red arrows).

group than in the control group by histological analyses ($1516.5 \pm 132.5$ vs. $689.9 \pm 70.6/mm^2$, $P < 0.001$; Fig 7B). Additionally, those GFP$^+$-cells were recruited along the border zone with SDF1 over-expression (Fig 7C). In contrast, significant recruitment of GFP$^+$ BM-cells was not observed at the remote area where significant expression of SDF1 was not observed.

## Discussion

In the first examination, we demonstrated that systemic administration of HMGB1 fragment could inhibit adverse LV remodeling by inducing angiogenesis and reducing fibrosis, leading to an improvement of LVEF. The first examination also showed that the number of MSCs at the damaged heart tissue site was significantly increased in after treatment with the HMGB1 fragment. In the second examination, we investigated the homing factor of MSCs in the damaged myocardium. Our results confirmed the significant expression of SDF1 around the peri-infarction area prior to systemic administration of HMGB1 fragment. The third examination using the GFP-BMT rat MI model showed that the recruitment of PDGFRα$^+$/CD90$^+$-BM cells to the peri-infarction area was further enhanced by the HMGB1 fragment. Moreover, confocal microscopic imaging revealed that the PDGFRα$^+$ BM-cells might release growth factors such as VEGF, and that some might have differentiated into vessel constituent cells at the damaged myocardium.

We investigated the mechanisms through which BM-derived mesenchymal cells including BM-MSCs are recruited to the infarcted myocardium in response to HMGB1 fragment administration. In this regard, we have previously reported that HMGB1 promotes PDGFRα$^+$ BM-MSCs aggregation around blood vessels in the bone marrow and that those BM-MSCs

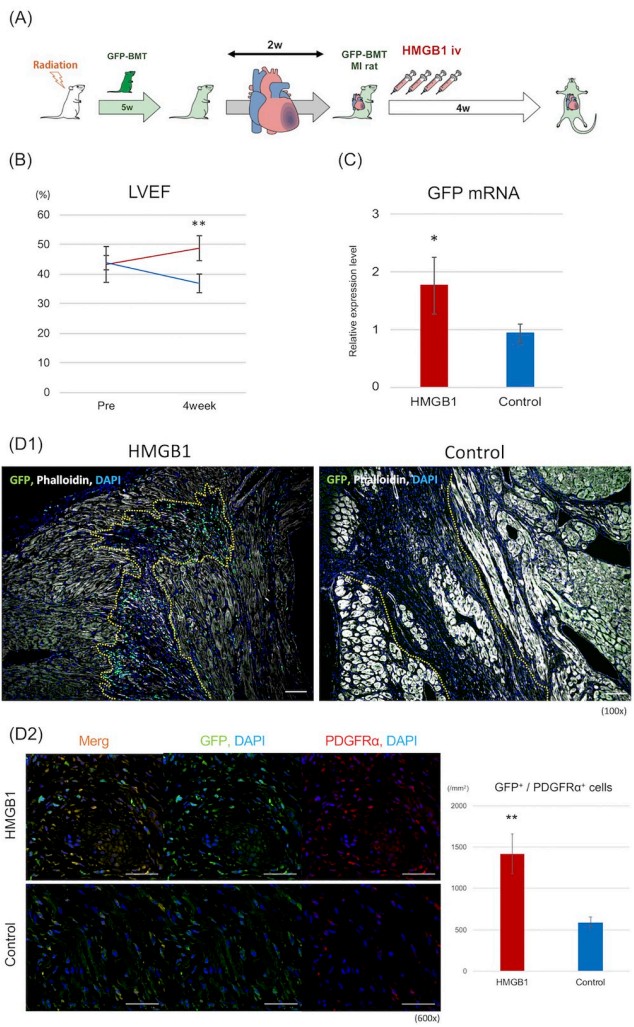

**Fig 5. Third examination using a green fluorescent protein bone marrow transplantation (GFP-BMT) rat model.** To investigate whether BM-MSCs were mobilized to the damaged myocardium after HMGB1 injection, a third examination using GFP-bone marrow transplantation (BMT) model rats was performed similarly to the first examination. A: Study protocol of the third examination. B: LVEF shows a significant improvement in the HMGB1 group (n = 8) compared to control (n = 7) after 4 weeks. C: On RT-PCR analysis, GFP expression in the border zone was significantly higher in the HMGB1 group than in the control. D: 1) Representative photomicrographs of the peri-infarction area after each treatment (100×, scale bar = 100 μm). Border lines are shown as yellow dotted lines. 2) Representative photomicrographs of PDGFRα staining (600×, scale bar = 50 μm). A larger number of GFP+/PDGFRα+ is observed in the HMGB1 group than in the control group. *P* -values were calculated using the Welch's t-test. $P < 0.05^*$, $P < 0.01^{**}$.

migrate via the circulatory system [12]. In another previous study, we have demonstrated that HMGB1 can induce the expression of C-X-C chemokine receptor 4 (CXCR4) on the surface of recruited BM-MSCs, both *in vivo* and *in vitro* [13]. The ligand of CXCR4 is SDF1, which plays an important role in the migration and proliferation of various stem cells, including BM-MSCs [19, 20].

In the present study, we revealed that SDF1 expression was significantly increased in MI rats, particularly in the peri-infarction area (Fig 3B and 3C), consistent with the findings of previous studies that showed that SDF1 expression increases in ischemic lesions, such as the microenvironment of tumors where fibroblasts, epithelial cells, or endothelial cells secrete

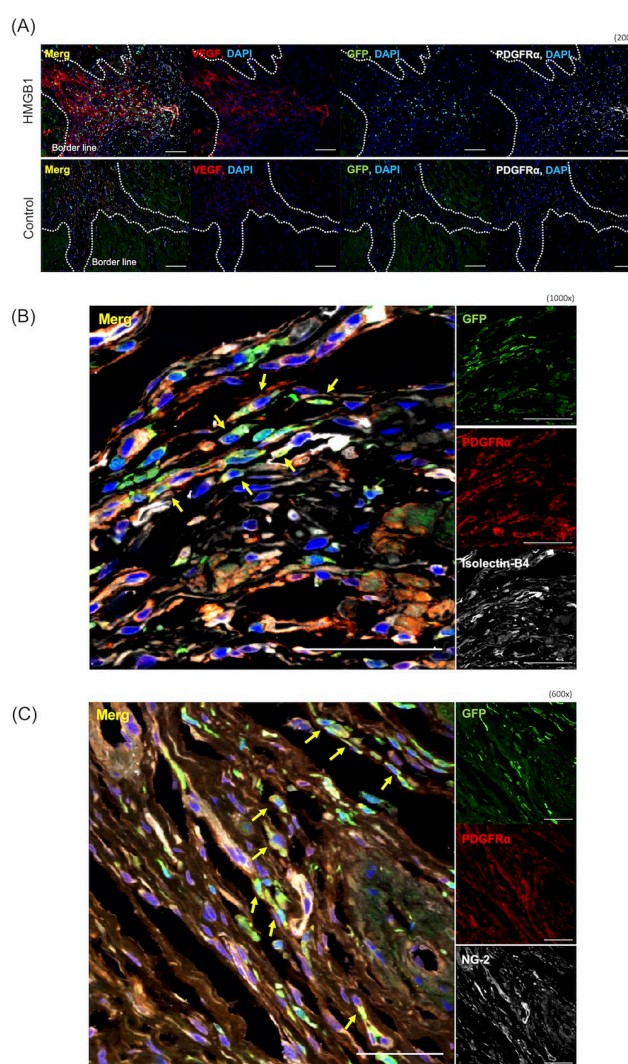

**Fig 6. Paracrine activity and differentiation of the recruited BM-MSCs in the damaged heart tissue.** A: Confocal laser microscopy analysis revealed VEGF-A expression around the GFP$^+$/PDGFR$\alpha^+$ cells (200×, scale bar = 100 μm). Each white dotted line depicts the border. B, C: Histological analysis of GFP$^+$/PDGFR$\alpha^+$ cells differentiating to vessel constituent cells in the peri-infarction area. In the HMGB1 group, some GFP$^+$/PDGFR$\alpha^+$ cells stained with anti-isolectin-B4 (B. 1000×, scale bar = 50 μm) or anti-NG-2 (C. 600×, scale bar = 50 μm) antibodies were observed in a subset of vessels within the peri-infarction area (yellow arrows).

SDF1 [19, 20, 23]. Given these findings, BM-MSCs might be mobilized to the peri-infarction area via CXCR4/SDF1 signaling complex (Fig 8). In our present study, the cell-cell junctions of vascular endothelial cells in the peri-infarction area were weakened. Theoretically, it is possible that BM-MSCs adhere to the walls of vessel in the border zone via CXCR4/SDF1 signaling, and accordingly migrate from the vessels to the ECM via the gaps between vascular endothelial cells (Fig 4).

We also investigated the mechanisms by which systemic injection of HMGB1 fragment inhibits adverse LV remodeling. We considered that several factors could be associated with the regenerative effects of the HMGB1 fragment. Similar to our study, several in vivo studies using HMGB1 have showed that HMGB1 can induce angiogenesis through VEGF in ischemic

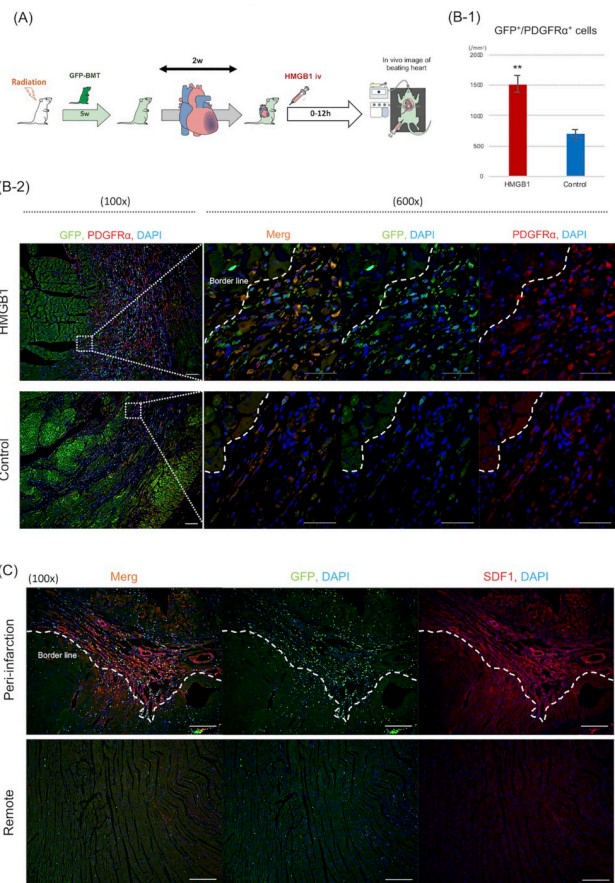

**Fig 7. Intravital imaging of GFP-BMT rat MI model after HMGB1 treatment.** Intravital imaging analysis was performed using the GFP-BMT rat MI model to visualize HMGB1-induced GFP⁺-cells mobilization to the damaged heart tissue in real time. A: Details of the study protocol using intravital imaging. B: Histological findings in GFP-BMT rat MI model 12 h after HMGB1 treatment. 1, 2) More GFP⁺/PDGFRα⁺ cells were visualized in the HMGB1 group compared with the control (100×, 600×; scale bar = 100, 200 μm, respectively). C: In the HMGB1 group, GFP⁺-cells were recruited along the peri-infarction area with SDF1 over-expression (100×, scale bar = 200 μm). Conversely, the recruitment of GFP⁺-cells was not enhanced at the remote area, where significant expression of SDF1 was not observed. *P* -values were calculated using the Welch's t-test. $P < 0.01^{**}$.

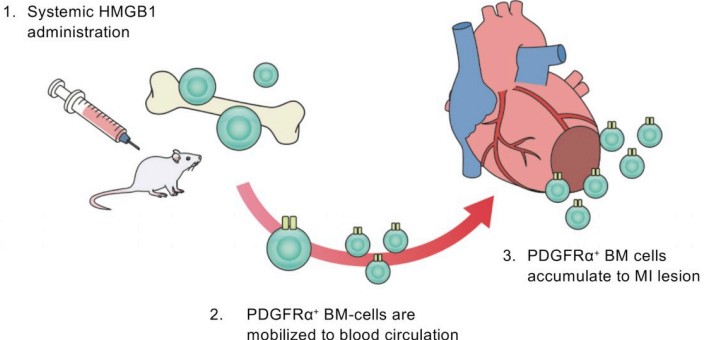

**Fig 8. Schematic outline of the present study.** Systemic administration of HMGB1 fragment can mobilize BM mesenchymal cells, including BM-MSCs, to blood circulation. Consequently, these BM-derived mesenchymal cells accumulate in the damaged myocardium through the SDF1/CXCR4 signaling complex, leading to functional recovery by paracrine activity of the BM-MSCs or to differentiation of some vascular constituent cells.

tissue, such as peripheral artery disease and acute MI [24–26]. Indeed, in the present study, a significant increase in *VEGF-A* expression was found in the peri-infarction area of the HMGB1 group as compared with the control (Fig 2G). The present study also revealed the overexpression of VEGF-A around the recruited BM-MSCs at the peri-infarction area (Fig 6A). Given these findings, we hypothesized that the HMGB1 fragment induced PDGFRα+ BM-cells, which secreted various growth factors such as VEGF in the damaged myocardium. We also hypothesized that paracrine activity of the recruited BM-MSCs could contribute to the therapeutic effects of HMGB1 fragment, as it has been reported that the regenerative effects of cell-based therapies using various stem cells can be attributed to paracrine activity [27–31]. This study also suggested that the recruited GFP+/PDGFRα+ cells may have differentiated into vessel constituent cells, such as vascular endothelial cells or pericytes, in the peri-infarcted area (Fig 6B and 6C). Differentiation of the recruited BM-MSCs may be a crucial factor associated with the regenerative effects of the HMGB1 fragment. Indeed, numerous *in vitro* and *in vivo* studies have revealed that BM-MSCs have the potential to differentiate into endothelial cells, pericytes, and smooth muscle cells [5, 32]. It is also conceivable that HMGB1 itself might have a regenerative effect in adverse LV remodeling. Takahashi and colleagues have reported that direct injection of HMGB1 into the myocardium can attenuate local myocardial inflammation, leading to the inhibition of cardiomyocyte hypertrophy and expansion of fibrosis [33]. With respect to adverse effect of full-length HMGB1 as inflammation mediator, the present study showed that the inflammatory activity in heart tissue might have been attenuated in MI rats treated by the HMGB1 fragment, given that we observed significant decreases in IL-1β and IL-6 levels after HMGB1 treatment (Fig 2G). Results from our previous study have demonstrated that CD68+ inflammatory cells in the heart tissue of a hamster model of dilated cardiomyopathy are significantly decreased in the HMGB1 treated group compared with control group [11], which suggests that the HMGB1 fragment could not enhance inflammation compared with the full-length HMGB1 (Fig 1).

There are essentially two types of cardiac regenerative therapy for ischemic cardiomyopathy (ICM), cell-based regenerative therapy (CBRT) [27] and endogenous regenerative therapy (ERT) [25, 26]. To date, CBRT has been the mainstream approach, and numerous studies have reported its usefulness with different types of stem cells [30, 31], including MSCs [8, 9]. CBRT requires an in-house cell-processing center with an aseptic environment in the hospital [34]. Because drug-induced regenerative therapy does not require any cell culture, the quality of those cells is more easily maintained than with CBRT. An additional limitation of CBRT is that these stem cells are usually introduced via several invasive delivery methods, including intra-coronary and intra-myocardial administration, and cellular sheet implantation [27–30]. In contrast, ERT with HMGB1 administration does not require those invasive procedures. Further clarification of the mechanism of CBRT would facilitate progress in regenerative drug discovery, thereby increasing the possibility of ERT offering a minimally invasive therapy for patients with ICM in the future.

This study should be carefully interpreted because of limitations such as the relatively small number of MI rats, especially in the second examination using the GFP-BMT rat MI model. The angiogenesis mechanism through VEGF-A by the recruited BM-MSCs should be further investigated. Because VEGF-A plays an important role in angiogenesis of HMGB1 as shown in our study and in previous reports [24–26], it would be useful to evaluate a model with blocked VEGF pathway to determine whether this increase has a causal role in documented angiogenesis. With respect to the mobilized BM-MSCs differentiation into the endothelial cells or pericytes, it is unclear whether GFP+/PDGFRα+ and IL-B4 or NG2 are co-expressed because PDGFRα seems to be more highly expressed in the vessels, although other cells also seem to be slightly stained in red. To determine more precisely that the recruited BM-MSCs can differentiate into these vessel constituent cells, more analyses will be required.

Moreover, the present study showed the regenerative effects of the HMGB1 fragment for MI in rats. To translate our results for application in the clinical setting, pre-clinical studies with larger animal models such as the porcine model of MI are required.

## Conclusion

Systemic administration of the HMGB1 fragment induces angiogenesis and reduces fibrosis by mobilizing BM-MSCs to the peri-infarction area, thereby providing a potential new approach for the treatment of ICM with CHF.

## Supporting information

**S1 Video. Intravital imaging in the GFP-bone marrow transplantation model.** Videos of the intravital imaging analysis performed in GFP-BMT rat MI model. Continuous intravital imaging with fixed view (0–12 h, 250×) showed that GFP$^+$-BM cells gradually accumulated in the damaged heart tissue after HMGB1 treatment.
(MP4)

**S1 Fig. Histological analysis of CD90$^+$/PDGFRα$^+$ cells 4 weeks after HMGB1 treatment.** CD90$^+$/PDGFRα$^+$ and DAPI$^+$ cells were counted as MSCs (A: 600×, scale bar = 50 μm, white arrows). Cells negative for CD90, PDGFRα, or DAPI were excluded (B: 600×, scale bar = 50 μm).
(TIFF)

**S1 File. All data of statistical analysis in the present study Examination 1.**
(DOCX)

## Acknowledgments

We thank Katsuto Tamai for preparing the HMGB1 fragment; Mami Nishida for performing BMT assays; and Yuri Ide, Hirohito Ayame, and Noriko Mochizuki-Oda for their excellent technical assistance.

## Author Contributions

**Conceptualization:** Shigeru Miyagawa, Yoshiki Sawa.

**Data curation:** Takasumi Goto.

**Formal analysis:** Takasumi Goto.

**Investigation:** Takasumi Goto.

**Methodology:** Takasumi Goto, Shigeru Miyagawa, Katsuto Tamai, Ryohei Matsuura, Takashi Kido, Toru Kuratani, Kazuo Shimamura, Ryoto Sakaniwa, Akima Harada, Yoshiki Sawa.

**Supervision:** Shigeru Miyagawa, Katsuto Tamai, Toru Kuratani, Yoshiki Sawa.

**Writing – original draft:** Takasumi Goto.

**Writing – review & editing:** Shigeru Miyagawa.

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
