## [Decision Letter · Decision Letter 0]

15 Jan 2020

PONE-D-19-32880

High-mobility group box 1 fragment suppresses adverse post-infarction remodeling by recruiting PDGFRα-positive bone marrow cells

PLOS ONE

Dear Dr. Takasumi Goto,

Thank you for submitting your manuscript to PLOS ONE. After careful consideration, we feel that it has merit but does not fully meet PLOS ONE’s publication criteria as it currently stands. Therefore, we invite you to submit a revised version of the manuscript that addresses the points raised during the review process.

Specifically, the reviewers raised criticisms concerning the quality of the images and asked for more detailed methods. Further, they also raised doubts on the induction of angiogenesis in the model.

We would appreciate receiving your revised manuscript by 3 months. To enhance the reproducibility of your results, we recommend that if applicable you deposit your laboratory protocols in protocols.io, where a protocol can be assigned its own identifier (DOI) such that it can be cited independently in the future. For instructions see: http://journals.plos.org/plosone/s/submission-guidelines#loc-laboratory-protocols

We look forward to receiving your revised manuscript.

Kind regards,

Federica Limana

Academic Editor

PLOS ONE

Journal Requirements:

Reviewers' comments:

Reviewer's Responses to Questions

**Comments to the Author**

1. Is the manuscript technically sound, and do the data support the conclusions?

Reviewer #1: Partly

Reviewer #2: Partly

2. Has the statistical analysis been performed appropriately and rigorously? 

Reviewer #1: Yes

Reviewer #2: I Don't Know

3. Have the authors made all data underlying the findings in their manuscript fully available?

Reviewer #1: Yes

Reviewer #2: Yes

4. Is the manuscript presented in an intelligible fashion and written in standard English?

Reviewer #1: Yes

Reviewer #2: Yes

5. Review Comments to the Author

Reviewer #1: 1. The authors found that HMGB1 induces angiogenesis in vivo. This has already been demonstrated in several works (which have not been mentioned, e.g.: Int J Cardiol. 2017 Dec 15;249:349-356 - Diabetes. 2010 Jun;59(6):1496-505 - J Mol Cell Cardiol. 2008 Apr;44(4):683-93). The additional data is interesting, but it cannot disregard what has already been documented.

2. The increase in VEGF could be an epiphenomenon. It would be useful to use a blocking model of the VEGF pathway (monoclonal antibodies, viral vectors, etc.) to verify whether this increase has a causal role in documented angiogenesis.

3. How can it be demonstrated that the injection of cells does not induce an inflammatory response, which alone is responsible for the increase in VEGF?

Reviewer #2: The ms by Takasumo Goto et al describe an interesting scenario, where administration of a short fragment of HMBG1 can help the recovery process after myocardial infarction. If the recruitment of PDGFRa+ cells from the bone marrow can be enhanced it can be of clinical relevance. However, the present manuscript is not ready for publication as it stands now. Major comments are that the whole manuscript is a bit unorganized, and the presented images are too small and at too low magnification to fully support the conclusions that the authors claim.

Specific comments:

It should be clear already in the abstract that it is a fragment of HMGB1 that is administered. Why is only the fragment (and not the full length protein administered)? Do they have different function or efficiency? This information should be presented already in the introduction.

The whole section on Material and methods would benefit from being re-written. As it is now, it is a mix of different sections – some describing specific methods and other describing the setup of the different experiments. It is confusing to read and also some information is missing. Explain better the procedures of the heart tissue, from fixation to image analysis. How long time was the hearts fixed in 10% formalin before immunohistochemistry? How were the hearts sectioned, and how thick sections?

Explain better how mRNA was measured. It is not enough to write “At the peri-infarction area, the level of GFP mRNA (Applied Biosystems) was evaluated” as in page 11, line 11.

Describe how the cardiomyocyte size were measured.

Clarify how the the different examinations are numbered. The text says number 1,2,3 and the figures say 1, 1-2 ,2 and 2-2.

Page 8-9:

Clarify at what age the evaluation of cardiac function was assessed. First it is written that it is done 2 weeks after ligation, and later it says 1 and 4 weeks after administration.

The study protocols (Fig. 2A, 3A, 4A, 5A) are nice, but they can be improved with better time line.

Page 13

Specify what primers were used for RT-PCR for the different genes. Occasionally, it says QT-PCR in the text…

Page 21, line 12

If there is no significant difference, it is wrong to say that LVDd is smaller. The sentence should be rephrased, eg. “There was no difference in LVDd between the HMGB1 group and the control group.” Also change in the figure legend.

The same goes for example TGFb, in Fig 2G.

Define which VEGF ligand that was analysed– both with RT-PCR and immunohistochemistry.

Results:

Fig.2

How many times was HMBG1 injected? In material&methods it says 1 time, but in Fig2A it looks as 4 injections are performed.

Fig. 2F

Include images in higher magnification to show the morphology of the CD90+/Pdgfra+ cells. In these images the colored pixels are irregular in shape and it seems as CD90 and PDGFRa in some cases show exactly the same shape. Is this expected?

Please, give an example in a photo what colored pixels that were counted as cells and which that were not.

Fig. 3D

How representative are the images with weakened cell-cell junctions? Can this be quantified? Have you tried to perform immunohistochemistry for any tight-junction markers?

Fig. 4D-1 Chose another color than white for the dotted lines in this image.

Several images are very small and with low magnification. It is difficult to interpret the individual cells. For example in Fig. 4D-2; Fig4E

Fig. 4D-2 show that not all GFP+ cells express Pdgfra. What kind of cells are GFP+/Pdgfra-?

Fig. 4F, G

It is not clear that GFP/PDGFRa and isolection/NG2 are coexpressed in certain cells, as none of the stainings are very specific. For example, the red color is more or less present I all cells. It is interesting to speculate that the cells differentiate into pericytes or endothelial cell, but more analyses are needed.

Video

Were the two samples identical from start, which they should have been. The HMBG1-sample has much more green cells both circulating and in the tissue when the video starts, compared to the sample that gets PBS.

Discussion

Page 31, line 3

A higher density of vessels was shown in the HMBG1 group, but it has not been shown that HMBG1 induce angiogenesis.

6. PLOS authors have the option to publish the peer review history of their article (what does this mean?). If published, this will include your full peer review and any attached files.

Reviewer #1: No

Reviewer #2: No

---

## [Author Response · Author response to Decision Letter 0]

24 Feb 2020

Responses to the Reviewers’ Comments

Review Comments to the Author

Reviewer 1: 

Comment 1. The authors found that HMGB1 induces angiogenesis in vivo. This has already been demonstrated in several works (which have not been mentioned, e.g.: Int J Cardiol. 2017 Dec 15;249:349-356 - Diabetes. 2010 Jun;59(6):1496-505 - J Mol Cell Cardiol. 2008 Apr;44(4):683-93). The additional data is interesting, but it cannot disregard what has already been documented.

Response 1: Thank you for your comments. As you mentioned, the angiogenetic ability of HMGB1 in ischemic tissue has already been reported. In those studies, HMGB1 treatment has been shown to induce VEGF over-expression in ischemic tissue, similar to the results in our present study, leading to neovascularization in the damaged tissue. In the present study, we also showed that HMGB1 can recruit PDGFRα+ mesenchymal stem cells (MSCs) from the bone marrow to the ischemic heart tissue, and that these recruited cells may release several therapeutic cytokines such as VEGF and can lead to angiogenesis. We added the references you mentioned related to the angiogenetic ability of HMGB1 to the revised Discussion section and the reference list (page 29, lines 11-14; page 39, lines 16-17; and page 40, lines 1-7).

Comment 2. The increase in VEGF could be an epiphenomenon. It would be useful to use a blocking model of the VEGF pathway (monoclonal antibodies, viral vectors, etc.) to verify whether this increase has a causal role in documented angiogenesis.

Response 2: Compared with control, VEGF expression in the HMGB1 group was significantly increased at the peri-infarction area where BM-MSCs were significantly mobilized. Several in vivo and in vitro studies have reported that BM-MSCs have the potential to release many therapeutic cytokines such as VEGF. However, as you mentioned, the increase in VEGF could be an epiphenomenon following myocardial infarction. We think that additional examination using a model of blocked VEGF pathway will be required to clarify the underlying mechanism by which the recruited BM-MSCs induce angiogenesis through VEGF. We added this information as a limitation of the present study to the revised manuscript (page 32, lines 4-8).

Comment 3: How can it be demonstrated that the injection of cells does not induce an inflammatory response, which alone is responsible for the increase in VEGF?

Response 3: In the present study, we administrated the HMGB1 fragment to MI model rats. The HMGB1 fragment is a peptide. In the first examination, mRNA levels of IL-1β and IL-6, which are representative inflammatory cytokines, were significantly higher in the control group than those in the HMGB1-treated group. In addition, in our previous study, we have reported that CD68+ inflammatory cells in the heart tissue of a hamster model of dilated cardiomyopathy are significantly decreased in the HMGB1-treated group compared with that in the control. Given these findings, the HMGB1 fragment could not enhance local inflammation in the damaged heart tissue. We added this discussion associated with inflammatory reaction after HMGB1 treatment to the revised Discussion section (page 30, lines 15-17; and page 31, lines 1-5).

Reviewer 2: 

The ms by Takasumo Goto et al describe an interesting scenario, where administration of a short fragment of HMBG1 can help the recovery process after myocardial infarction. If the recruitment of PDGFRa+ cells from the bone marrow can be enhanced it can be of clinical relevance. However, the present manuscript is not ready for publication as it stands now. Major comments are that the whole manuscript is a bit unorganized, and the presented images are too small and at too low magnification to fully support the conclusions that the authors claim.

Specific comments:

Comment 1: It should be clear already in the abstract that it is a fragment of HMGB1 that is administered. Why is only the fragment (and not the full length protein administered)? Do they have different function or efficiency? This information should be presented already in the introduction.

Response 1: 

 Thank you for all your comments. It is well known that full-length HMGB1 has two different effects. First, the inflammatory reaction. In acute inflammation, HMGB1 works as mediator of inflammation. Previous studies have already shown that the TLR-2, -4, or RAGE-binding domains of HMGB1 are associated with systemic inflammation. 

 On the other hand, HMGB1 is reportedly also a regenerative factor to repair damaged tissue. Our previous studies have shown that HMGB1 mobilizes CXCR4+-BM-MSCs from the bone marrow to the damaged skin. We also demonstrated that the recruited BM-MSCs promote skin repair, and that some BM-MSCs can differentiate into skin constituent cells.

 Inflammatory reaction can enhance adverse remodeling in various heart diseases. Focusing on this regenerative effect of HMGB1, we removed the HMGB1 domains described as being related to systemic inflammation. Thus, the fragment of HMGB1 was obtained. In our previous study using a hamster DCM model, we found that this HMGB1 fragment inhibits fibrosis relative to the control. In addition, we have reported that CD68+ inflammatory cells in the heart tissue of the DCM model are significantly decreased in the HMGB1-treated group compared with that in the control.

 Similar to our previous studies, our present study demonstrates that this HMGB1 fragment inhibits post-MI adverse remodeling such as fibrosis and cardiomyocyte hypertrophy, and that it does not enhance the inflammatory activity in the damaged heart tissue. We added this information to the revised manuscript (page 6, lines 6-11).

Comment 2: The whole section on Material and methods would benefit from being re-written. (a) As it is now, it is a mix of different sections – some describing specific methods and other describing the setup of the different experiments. It is confusing to read and also some information is missing. (b) Explain better the procedures of the heart tissue, from fixation to image analysis. (c) How long time was the hearts fixed in 10% formalin before immunohistochemistry? (d) How were the hearts sectioned, and how thick sections?

(e) Explain better how mRNA was measured. It is not enough to write “At the peri-infarction area, the level of GFP mRNA (Applied Biosystems) was evaluated” as in page 11, line 11. 

(f) Describe how the cardiomyocyte size were measured.

Response 2: Thank you for your comments concerning the Material and methods section. As you pointed out, we felt that this section would be a little confusing to read because some information overlapped with study protocols and each analysis. 

(a) Regarding the Material and methods section, we changed the order of paragraphs as follows: Animal care, Short length of HMGB1 fragment, Study protocol of each examination (Examination 1-3, Intravital imaging), Echocardiography, Histological analysis, Real-time (RT) PCR analysis, and Statistical analysis (page 7, line 1, to page 15, line 7). 

(b) In the revised Histological analysis subsection, we explain the fixation of the heart sample, and subsequently mentioned how to evaluate each histological image in each examination (page 12, line 8, to page 14, line 6).

(c) All the excised heart samples were fixed with 10% buffered formalin for paraffin-embedded sections or with 4% paraformaldehyde for frozen sections for over a day. We mention this in the revised Histological analysis subsection (page 12, lines 10-12).

(d) The heart was resected perpendicularly to the long axis of the left ventricle in slices measuring a few mm. We added this information to the revised Histological analysis subsection (page 12, lines 9-10).

(e) We modified the study protocol of examination 3 and excluded the GFP primer information from the protocol. Instead, we added the information on GFP primers to the revised RT-PCR analysis subsection, following your suggestion (page 14, lines 13-15).

(f) The paraffin-embedded sections were stained with periodic acid-Schiff to assess cardiomyocyte hypertrophy in each group. Using light microscopy, the short-axis diameter of myocytes was counted in 10 randomly selected fields, and the average number was calculated. We added how cardiomyocyte hypertrophy was assessed to the revised Histological analysis subsection (page 12, lines 15-17; and page 13, lines 1-2).

Comment 3: Clarify how the different examinations are numbered. The text says number 1,2,3 and the figures say 1, 1-2 ,2 and 2-2.

Response 3: I apologize for the wrong numbering of examinations in the submitted Figures. The numbering of examinations in the main text is correct. Examination 1 was aimed at evaluating the regenerative effects of HMGB1 in an MI model rat. In examination 2, we investigated SDF1 expression, which is a representative homing factor of BM-MSCs, in the damaged heart tissue prior to HMGB1 treatment. Examination 3 was performed to clarify the enhanced mobilization of BM-MSCs to the damaged heart tissue after HMGB1 treatment using a GFP-bone marrow transplantation (GFP-BMT) rat MI model. Additionally, intravital imaging analysis with the same GFP-BMT rat MI model was performed to investigate the recruitment of BM cells to damaged heart tissue after HMGB1 treatment in real time. We corrected the numbering of examinations in the revised manuscript (new Figures 2, 3, 5, and 7).

Comment 4: Page 8-9: Clarify at what age the evaluation of cardiac function was assessed. First it is written that it is done 2 weeks after ligation, and later it says 1 and 4 weeks after administration. The study protocols (Fig. 2A, 3A, 4A, 5A) are nice, but they can be improved with better time line.

Response 4: In all examinations of the present study, we used MI model rats 2 weeks after ligation of the left coronary artery. In examination 1, we administered HMGB1 to these MI model rats. We evaluated cardiac function in each group before HMGB1 or PBS administration, and reassessed cardiac function at 1 and 4 weeks after HMGB1 injection. We modified this section in the revised manuscript (page 12, lines 3-5). Additionally, as you suggested, we added a timeline in each figure (new Figure 2A, 3A. 5A, and 7A).

Comment 5: Page 13: Specify what primers were used for RT-PCR for the different genes. Occasionally, it says QT-PCR in the text…

Response 5: We have added the assay IDs for all the primers used in the present study (page 14, lines 11-14). We performed real-time PCR (RT-PCR) analysis in the present study. We changed "QT-PCR" to "RT-PCR" in the revised manuscript.

Comment 6: Page 21, line 12: If there is no significant difference, it is wrong to say that LVDd is smaller. The sentence should be rephrased, eg. “There was no difference in LVDd between the HMGB1 group and the control group.” Also change in the figure legend. The same goes for example TGFb, in Fig 2G.

Response 6: As you mentioned, we considered that any result with p-values over 0.05 should not be mentioned as being higher or lower compared with the control. We therefore excluded the following sentences from the revised manuscript and figure legends:

・LVDd was smaller in the HMGB1 group, but not significantly different.

・LVDd was shorter in the HMGB1 group than in the control, but not significantly. 

・ ...whereas that of the septal area was lower in the HMGB1 group, although the difference was not significant (0.76 ± 0.12 vs. 0.83 ± 0.22, P = 0.37). 

Comment 7: Define which VEGF ligand that was analysed– both with RT-PCR and immunohistochemistry.

Response 7: In both RT-PCR and immuno-histological analyses, the VEGF ligand was VEGF-A. We revised this point.

Comment 8: Fig.2 

How many times was HMBG1 injected? In material & methods it says 1 time, but in Fig2A it looks as 4 injections are performed.

Response 8: 

 We apologize for the lack of this information in the submitted manuscript. Referring to our previous protocol, in examinations 1 and 3, HMGB1 (3 mg/kg/day) was administered for 4 days. We added this information to the revised manuscript (page 8, lines 11-14, and page 10, line 3). 

Comment 9: Fig. 2F Include images in higher magnification to show the morphology of the CD90+/Pdgfra+ cells. In these images the colored pixels are irregular in shape and it seems as CD90 and PDGFRa in some cases show exactly the same shape. Is this expected?

Please, give an example in a photo what colored pixels that were counted as cells and which that were not.

Response 9: In the first examination, we counted CD90+/PDGFRα+/DAPI+ cells as MSCs, and we excluded cells that were negative for CD90, PDGFRα, or DAPI. 

 As you mentioned, the submitted figures were slightly unclear; therefore, we changed Figure 2F from the previous image at 400× magnification to a clearer image at 600× to improve the resolution (New Figure 2F). Furthermore, we have added Supplemental Figure 1, which is a representative photo showing which colored pixels were counted as CD90+/PDGFRα+ cells and which were not (Supplemental Figure 1). 

Comment 10: Fig. 3D

How representative are the images with weakened cell-cell junctions? Can this be quantified? Have you tried to perform immunohistochemistry for any tight-junction markers?

Response 10: In the infarcted myocardium, many cell-cell junctions in endovascular cells were weakened, with tight junctions unclear in the lesion (new Figure 4A). Furthermore, some of them were completely disrupted (New Figure 4B). In contrast, in normal heart tissue, tight junctions between individual endothelial cells were more clearly observed compared with that in the damaged heart tissue (new Figure 4C, 4D). However, quantifying the damage to cell-cell junctions was difficult. 

As you mentioned, prior to the above electron microscopy analysis, we evaluated the expression of VE-cadherin at cell-cell junction in the damaged heart tissue using immuno-histological analysis. However, the expression of VE-cadherin at cell-cell junctions was unclear in all images, therefore we evaluated the cell-cell junction by electron microscopy analysis. Thus, we added several electron microscopy images with several magnifications to the revised manuscript (new Figure 4A, 4B, 4C, 4D). 

Comment 11: Fig. 4D-1

Chose another color than white for the dotted lines in this image. 

Response 11: Following your suggestion, we changed the color of the dotted lines from white to yellow (page 24, line 1, and new Figure 5D-1).

Comment 12: Several images are very small and with low magnification. It is difficult to interpret the individual cells. For example in Fig. 4D-2; Fig4E.

Response 12: Thank you for your comments. For the initial submission, it was difficult for us to incorporate all the figures associated with examination 3 into only one page. Following your suggestion, we enlarged Figure 4D-2 and made the new Figure 5 (new Figure 5D-2). We separated Figure 4E, 4F, and 4G from Figure 4, and made them the new Figure 6 in the revised manuscript (new Figure 6).

Comment 13: Fig. 4D-2 show that not all GFP+ cells express Pdgfra. What kind of cells are GFP+/Pdgfrα-?

Response 13: In our other study investigating the effects of the HMGB1 fragment on bone marrow, we confirmed that HMGB1 led to the mobilization of not only BM-MSCs but also of endothelial progenitor cells (EPCs) from the bone marrow to the blood circulation. In the present study, we considered that the GFP+/PDGFRα+ cells were BM-MSCs as we had performed total GFP+-bone marrow transplantation. In contrast, we assumed that GFP+/ PDGFRα- cells were leukocytes such as M2 macrophages or BM-derived vascular endothelial cells such as EPCs.

Comment 14: Fig. 4F, G

It is not clear that GFP/PDGFRa and isolection/NG2 are coexpressed in certain cells, as none of the stainings are very specific. For example, the red color is more or less present I all cells. It is interesting to speculate that the cells differentiate into pericytes or endothelial cell, but more analyses are needed.

Response 14: In examination 3, we found that some GFP+/PDGFRα+ cells were present at the vessel component in the peri-infarction area after HMGB1 treatment. Because we used GFP-bone marrow-transplanted-rats, GFP+ cells were bone marrow-derived. Given this finding, we hypothesized that BM-MSCs would be mobilized to the damaged heart tissue by HMGB1, and that BM-MSCs might differentiate into vessel constituent cells.

 In these Figures (old Figure 4F and 4G), PDGFRα is colored red. PDGFRα is a BM-MSC marker in the bone marrow, and is also a representative marker of mesenchymal cells such as vascular endothelial cells and pericytes. PDGFRα seems to be more highly expressed in the vessels, although other cells also seemed to be stained slightly red, as you mentioned. This is a limitation of immuno-histological analysis. We mentioned this point in the Discussion section (page 32, lines 8-13).

Comment 15: Video

Were the two samples identical from start, which they should have been. The HMBG1-sample has much more green cells both circulating and in the tissue when the video starts, compared to the sample that gets PBS.

Response 15: The purpose of the intravital imaging analysis was to investigate whether the recruitment of GFP+ cells to the damaged heart tissue could be enhanced by systemic administration of HMGB1. In the video of the HMGB1 group, it can be seen that GFP+ cells are mobilized to the damaged heart tissue and gradually increased in number after HMGB1 treatment. In contrast, in the video of the control (PBS) group, the recruitment of GFP+ cells to the damaged heart tissue was not enhanced after PBS injection. In addition, after intravital imaging, we also evaluated GFP+/PDGFRα+ cells at the peri-infarction area in the HMGB1 and PBS groups using immuno-histological analysis. However, as you mentioned, in the preoperative video of the control group, GFP+ cells seem to be present in lower numbers than that in the HMGB1 group.

Therefore, we excluded the movie of the PBS group from the revised manuscript, as the enhancement of GFP cell recruitment after HMGB1 injection can be observed in the video of the HMGB1 group.

Comment 16: Discussion, Page 31, line 3

A higher density of vessels was shown in the HMBG1 group, but it has not been shown that HMBG1 induce angiogenesis.

Response 16: The angiogenetic ability of HMGB1 in the ischemic tissue has already been reported in several models (references No. 11-13 and 24-26). In those studies, HMGB1 treatment induces VEGF over-expression in ischemic tissue, consequently leading to neovascularization in the damaged tissue. Similar to those reports, the present study showed enhancement of capillary density and VEGF-A expression in the damaged heart tissue after HMGB1 treatment. In addition, our study demonstrated that the mobilization of BM-MSCs to the peri-infarction area was enhanced by systemic administration of HMGB1, and that VEGF-A was expressed around the recruited BM-MSCs in the damaged heart tissue. Given previous lines of evidence and our findings, we considered that HMGB1 can lead to angiogenesis in the damaged heart tissue through VEGF-A via the paracrine effects of the recruited BM-MSCs. However, as you mentioned, the present study did not show that HMGB1 induces angiogenesis directly. To clarify the mechanism through which HMGB1 affects angiogenesis via VEGF-A, additional examinations using a model of blocked VEGF pathway will be required. We mentioned this information as a limitation of the present study in the revised manuscript (page 32, lines 4-8).

---

## [Decision Letter · Decision Letter 1]

28 Feb 2020

High-mobility group box 1 fragment suppresses adverse post-infarction remodeling by recruiting PDGFRα-positive bone marrow cells

PONE-D-19-32880R1

Dear Dr. Takasumi Goto,

We are pleased to inform you that your manuscript has been judged scientifically suitable for publication and will be formally accepted for publication once it complies with all outstanding technical requirements.

With kind regards,

Federica Limana

Academic Editor

PLOS ONE

Additional Editor Comments (optional):

Reviewers' comments:

Reviewer's Responses to Questions

**Comments to the Author**

1. If the authors have adequately addressed your comments raised in a previous round of review and you feel that this manuscript is now acceptable for publication, you may indicate that here to bypass the “Comments to the Author” section, enter your conflict of interest statement in the “Confidential to Editor” section, and submit your "Accept" recommendation.

Reviewer #1: All comments have been addressed

2. Is the manuscript technically sound, and do the data support the conclusions?

Reviewer #1: Yes

3. Has the statistical analysis been performed appropriately and rigorously? 

Reviewer #1: Yes

4. Have the authors made all data underlying the findings in their manuscript fully available?

Reviewer #1: Yes

5. Is the manuscript presented in an intelligible fashion and written in standard English?

Reviewer #1: Yes

6. Review Comments to the Author

Reviewer #1: All comments have been addressed.

The article is very interesting and worthy of publication.

Accept.

7. PLOS authors have the option to publish the peer review history of their article (what does this mean?). If published, this will include your full peer review and any attached files.

Reviewer #1: No

---

## [Editor Report · Acceptance letter]

20 Mar 2020

PONE-D-19-32880R1 

High-mobility group box 1 fragment suppresses adverse post-infarction remodeling by recruiting PDGFRα-positive bone marrow cells 

Dear Dr. Goto:

I am pleased to inform you that your manuscript has been deemed suitable for publication in PLOS ONE. Congratulations! Your manuscript is now with our production department. 

With kind regards,

on behalf of

Dr. Federica Limana 

Academic Editor

PLOS ONE